# Association of Depression and Anxiety with Social Network Types: Results from a Community Cohort Study

**DOI:** 10.3390/ijerph18116120

**Published:** 2021-06-06

**Authors:** Saju Madavanakadu Devassy, Lorane Scaria, Natania Cheguvera, Kiran Thampi

**Affiliations:** 1Rajagiri College of Social Sciences (Autonomous), Kerala 683 104, India; lorane@rajagiri.edu (L.S.); nataniamicheal@gmail.com (N.C.); kiranthampis@rajagiri.edu (K.T.); 2Rajagiri International Centre for Consortium Research in Social Care (ICRS), Kerala 683 104, India

**Keywords:** depression, anxiety, social network types, mental health, India

## Abstract

Social networks protect individuals from mental health conditions of depression and anxiety. The association between each social network type and its mental health implications in the Indian population remains unclear. The study aims to determine the association of depression and anxiety with different social network types in the participants of a community cohort. We conducted a cross-sectional household survey among people aged ≥30 years in geographically defined catchment areas of Kerala, India. We used cross-culturally validated assessment tools to measure depression, anxiety and social networks. An educated male belonging to higher income quartiles, without any disability, within a family dependent network has lower odds of depression and anxiety. Furthermore, 28, 26.8, 25.7, 9.8 and 9.7% of participants belonged to private restricted, locally integrated, wider community-focused, family-dependent and locally self-contained networks, respectively. Close ties with family, neighbours, and community had significantly lower odds of anxiety and depression than private restricted networks. The clustering of people to each social network type and its associated mental health conditions can inform social network-based public health interventions to optimize positive health outcomes in the community cohort.

## 1. Introduction

In India, 197 million people have reported a prevalence of mental disorders. Of these, 45.7 million had depressive disorders, and 44.9 million had anxiety disorders [1]. A study examining the global burden of disease (GBD) estimated that the prevalence of anxiety and depression in India was 3.3% (3–3.5%) and 3.3% (3.1–3.6%), respectively [1]. The World Health Organization (WHO) predicted that by 2020, approximately 20% of Indians would develop mental illnesses [2]. Furthermore, a study examining GBD reported that among all Indian states, Kerala had the highest prevalence of both depression and anxiety. Mental health conditions, such as depression and anxiety, among populations worldwide are affected by social networks. Studies report an undeniable association between the mental health of people and the type of social network they belong to [3,4,5,6].

### 1.1. Social Networks as Protective Factors for Mental Health

Social networks are the systems of social relationships in which an individual embeds [7,8]. An individual’s social network consists of people with whom they regularly interact, the characteristics of such individuals and the quality of interactions. A support network is defined as the availability and accessibility to formal and informal platforms of social interactions. Individuals with limited support networks have lower levels of mental health and psychological wellbeing [3]. The feeling of being loved and cared for was associated with lower levels of anxiety, depression and somatization, and better adaptation to stressful situations [9]. Strong support networks can reduce the prevalence of mental health conditions. Support networks can provide physical and psychological advantages to people who encounter stressful events and reduce psychological distress [10]. Social integration is the extent to which individuals participate in various social relationships, including engagement in social activities or relationships, and a sense of commonality and identification with one’s social roles [11]. Low social integration is associated with poor health outcomes, including those related to mental health.

The present study is part of a more extensive cohort study called SWADES (social wellbeing and determinants of health study) [12] and aimed to determine depression and anxiety with support network types in participants included in the cohort. This study adopted Wenger’s concept of social network typology [13], which includes five distinct support network types. Each type represents different levels of integration into society, and the extent of support received within the social network. 

### 1.2. Social Network Typologies

The five social network types are as follows: family-dependent, locally integrated, locally self-contained, wider community-focused and private restricted. These social network types bases on the following criteria: proximity to close kin;proportion of family, friends, and neighbours involved;levels of interactions between people and their families, friends, neighbours, and community groups.

Wenger reported that individuals in family-dependent or private restricted support networks have the highest risk of loneliness, depression and other mental health conditions. In contrast, those in a locally integrated network have a lower risk of adverse mental health outcomes [14]. Furthermore, people in private restricted or locally self-contained networks had a higher risk of isolation than those in locally integrated or wider community-focused support networks [14]. 

Various studies have demonstrated a causal relationship between mental health and social network typologies, wherein social networks protect individuals from mental health issues such as depression [15,16,17,18,19]. Although studies have extensively examined the association between mental health and social networks, we used Wenger’s social network typology in this study to determine support networks among individuals aged ≥30 years. Previous studies conducted in India have used Wenger’s social network typology only in older adults. Studies have rarely explored the association between each social network type and its mental health implications in Indian individuals aged ≥30 years. This study will follow up the cohort over ten years to determine whether social networks change with changes in individual profiles in India and how these changes modulate participants’ mental health outcomes.

## 2. Materials and Methods

We conducted a cross-sectional household survey among people aged ≥30 years. The findings of this study used data from the SWADES, a more extensive cohort study in Keezhmad panchayat located in the Vazhakulam block of Ernakulam district, Kerala. We interviewed a total of 997 participants in this survey at their homes. 

We selected Keezhumadu Panchayath as the geographical location for data collection as it represents a mixed culture and a cross-section of socioeconomic characteristics of people in Kerala. We did a mapping exercise to geographically mark the boundaries and locate all the households with a participant aged above 30. The study is part of the SWADES cohort study, focusing on chronic conditions and mental health comorbidities in communities. For the study, we included inclusion criteria of 30 years and above for the participants, as chronic conditions and other comorbidities are usually seen among people aged 30 and above [20]. Trained postgraduate students conducted door to door surveys to identify the outcome variables through a structured questionnaire comprising standardized scales. All the questions were administered in the local language, Malayalam, by the students. Students obtained informed consent from all the participants before participation. A study protocol previously published [12] describes the recruitment of sample and further procedures in detail.

### 2.1. Measures

#### 2.1.1. Outcome Variables

The two primary outcomes measured were depression and anxiety. We assessed depression and anxiety using the Depression, Anxiety and Stress Scale (DASS), a self-report instrument [21]. DASS-42 is a set of three scales that separately measure depression, anxiety and stress. Each item evaluates the frequency of depressive or anxiety symptoms during the past week on a 4-point scale, with a score of 0, 1, 2, and 3 indicating rarely or none of the time, some of the time, much of the time, and most or all the time, respectively.

Scores for depression and anxiety were calculated by summing scores obtained for relevant items. Based on summative scores, the depression level of participants was categorised as normal (0–9), mild (10–13), moderate (14–20), severe (21–27), and extreme (28+). Similarly, anxiety was categorised as normal (0–7), mild (8–9), moderate (10–14), severe (15–19), and extreme (20+). This study considered an individual to have depression or anxiety if the DASS score was moderate or above. The cut-off scores for anxiety and depression determined using the DASS scale have been validated previously [20]. Studies conducted across India on common mental health conditions have used the DASS as a valid and reliable measure of depression, anxiety and stress. Cronbach internal consistency of the entire scale was 0.89. [22,23,24]. 

#### 2.1.2. Social Network Variables

The Practitioner Assessment of Network Type (PANT) described by Wenger was employed to ascertain the social network types of participants [8,13]. The construct validity of Wenger’s social network typology was tested and validated in an Indian setting [14]. The questionnaire includes the following eight questions: How far away does your nearest relative live? 0 (no relatives); 1 (with 1.6 km/same home); 2 (1.5–9.6 km); 3 (9.7–25.7 km); 4 (25.8–80.5 km); 5 (above 80.5 km)Where does your nearest sister or brother live? 0 (no siblings); 1(with 1.6 km/same home); 2 (1.5–9.6 km); 3 (9.7–25.7 km); 4 (25.8–80.5 km); 5 (above 80.5 km)Where does your nearest child live? 0 (no child); 1(with 1.6 km/same home); 2 (1.5–9.6 km); 3 (9.7–25.7 km); 4 (25.8–80.5 km); 5 (above 80.5 km)How often do you see any of your children or relatives to speak to? 0 (never); 1 (daily); 2 (2–3 times a week); 3 (at least weekly); 4 (at least monthly); 5 (less often)How often do you have a chat or do something with one of your friends? 0 (never); 1 (daily); 2 (2–3 times a week); 3 (at least weekly); 4 (at least monthly); 5 (less often)How often do you see any of your neighbours to have a chat or do something with? 0 (never); 1 (daily); 2 (2–3 times a week); 3 (at least weekly); 4 (at least monthly); 5 (less often)Do you attend religious meetings/visit religious places? 0 (no); 1 (yes, regularly); 2 (yes, occasionally)Do you attend meetings of any community or groups such as clubs, lectures or anything like that? 0 (no); 1 (yes, regularly); 2 (yes, occasionally)

Five network typologies, namely family-dependent, locally integrated, locally self-contained, wider community-focused and private restricted types, are described based on participant responses to the above questions. The five network types in detail include:A family-dependent network type is a network type predominantly characterised by (a) a relative living within a 1.6 km radius; (b) a child and/or sibling living within a radius of 9.6 km; (c) frequent contact with a relative (daily or 2–3 times a week); (d) limited contacts with friends or neighbours (never, less often or at least monthly); (e) and occasional attendance to religious meeting and community clubs.A locally integrated network type is characterised by (a) close proximity with a relative (living within the same house, within a radius of 9.6 km and/or within 25.7 km); (b) a child and/or sibling living close (within the same house, within a radius 9.6 km and/or a radius of 25.7 km); (c) very frequent contact with a relative (daily and/or 2–3 times a week); (d) frequent contacts with neighbours and/or friends (daily, 2–3 times a week and/or at least weekly); (e) regular attendance at religious meetings and community clubs.A locally self-contained network type is characterised by (a) an arm’s length proximity with relatives (living within a radius of 25.7 km and/or a radius of 80.5 km), children (no child, a child living within a radius of 25.7 km and/or a radius of 80.5 km) and siblings (living within a radius of 9.6 km and/or 25.7 km and/or 80.5 km); (b) occasional contact with relatives (at least weekly and/or at least monthly) and neighbours (at least weekly and/or at least monthly); (c) limited contact with friends (less often and/or at least monthly) and (d) occasional attendance at religious meetings and/or community clubs.A wider community focused network type is network type predominantly characterised by (a) absence of a relative (living within a radius of 80.5 km or above), child (living within a radius of 80.5 km or above or a sibling (living within a radius of 80.5 km or above; (b) limited contacts with relatives (less often and/or at least monthly); (c) occasional contacts with neighbours (at least weekly and/or at least monthly); (d) very frequent contact with friends (daily, 2–3 times a week and/or at least weekly) and regular attendance at clubs and religious meetings.A private restricted network is characterised by (a) lack of relatives (living within a radius of 80.5 km or above), child (living within a radius of 80.5 km or above) or siblings (living within a radius of 80.5 km or above); (b) minimal contact with relatives, friends or neighbours (less often) and (c) no involvement in religious meetings or clubs.

Physical proximity and frequency of communication are two important factors used by Wenger to identify the social networks in the PANT Social network questionnaire. Proximity is a fundamental way for people to connect, as a person tends to approach someone close to them rather than someone distant in the hour of need [25]. This is especially true in India, where the likelihood of travelling considerable distances are limited due to several reasons including limited public/private transport, poor and narrow roads, heavy traffic or rising fuel costs [26]. The exact algorithm used to construct the network variable has been described in detail in another study [13].

Private restricted social network characterised by little involvement in activities or relationships are identified as nonintegrated social networks. In contrast, social networks with good relationships and/or involvement in activities are identified as integrated social networks. 

#### 2.1.3. Sociodemographic Variables

We used a questionnaire to collect the following sociodemographic information of participants: age, sex, education, occupation and income. Chronic diseases (stroke, diabetes, hypertension, heart disease, tuberculosis, malaria and others) were self-reported.

#### 2.1.4. Disability

Disability was measured using the WHO Disability Assessment Schedule (DAS) 2.0 [27]. This instrument includes 12 questions scored on a four-point scale from 0 (no difficulty) to 4 (extreme difficulty/cannot do). Summative scores of the scale were categorised into four equal quartiles, and participants belonging to the last quartile were categorised as functionally disabled. The WHO-DAS was previously used and validated in a study conducted in India [28].

### 2.2. Data Analysis 

All statistical analyses were performed using SPSS (IBM Version 25, New York, NY, USA) and STATA (StataCorp LLC Version 15, Lakeway Drive, TX, USA). Descriptive analysis of the sociodemographic variables was performed using different social network typologies. The chi-square test and one way ANOVA was used to examine and find the differences of each social networks with the categorical and continuous demographic variables, respectively. Logistic regression was performed to analyze the association between factors and outcomes, and the odds ratio with a 95% confidence interval (CI) was calculated. Depression and anxiety scores were computed to estimate the prevalence of common mental health conditions among people belonging to different social network types. 

### 2.3. Ethical Considerations

Signed informed consent was obtained from all the participants. The authors assert that all procedures contributing to this work comply with the ethical standards of the relevant national and institutional committees on human experimentation and with the Helsinki Declaration of 1975, as revised in 2008. Ethical approval was obtained from (Anonymous) Hospital Institutional Ethics Committee (Registration No. ECR/1153/Inst/KL/2018, Study Reference Number: RAJH 18003).

## 3. Results

### 3.1. Sociodemographic Characteristics

A total of 997 participants aged ≥30 years were included in the analysis. Table 1 lists the sociodemographic characteristics of the study sample. The majority of participants in the sample were women (63.4%). 22.9% of the respondents were aged between 60 and 69 years, 42.8% were Muslim, 31.2% had completed primary education and 41.8% of the respondents belonged to the lowest income group. The mean age of participants was 53.9 (standard deviation [SD] = 14.2) years. Mean scores of depression and anxiety in the current study were 7.4 (s.d = 7.3) and 5.7 (s.d = 6.7). 

The study revealed that 28% of the participants belonged to the private restricted network type, 26.8% to the locally integrated, 25.7% belonged to the wider community-focused network type, 9.8% belonged to the family-dependent network type, and 9.7% belonged to a local self-contained network type. Table 1 presents the characteristics of each social network type. 

### 3.2. Social Network Typologies: Association with Depression

The overall prevalence of moderate and above depression was 15.7%, as determined based on DASS scores for depression. We observed that 23.3% of participants with a private restricted network exhibited depressive symptoms, whereas only 9.2% and 14.6% of participants within family-dependent and locally integrated social networks exhibited depressive symptoms (Table 1). The findings of logistic regression analysis revealed that the odds of depression were significantly lower in participants with family-dependent, locally integrated, locally self-contained and wider community-focused networks than in those with a private restricted social network. Furthermore, the odds of depression were lower in participants with a family-dependent network than in those with a community-focused network. (Table 2). 

Apart from social network typologies, age, sex, income, disability and the presence of multiple morbidities were associated with depression scores. For instance, male respondents were less likely to have higher depression scores when compared to women. Respondents with tertiary education had significantly lower odds of depression scores when compared to people who were uneducated. Findings also showed that higher levels of depression were nine times more likely among participants with disability than those without disability. In addition, higher levels of depression were four times more likely among participants with three or more chronic conditions compared with those without a diagnosed chronic condition (Table 2). 

### 3.3. Social Network Typologies: Association with Anxiety

According to DASS scores calculated for anxiety, the overall prevalence of moderate and above anxiety scores in the study sample was 21.5%. We observed that 32.6% of participants with a private restricted social network exhibited moderate and above anxiety scores, whereas only 16.3% and 16.1% of participants with family-dependent and locally integrated social networks exhibited moderate and above anxiety scores, respectively. Table 3 described logistic regression analysis of anxiety with different variables.

The findings of the logistic regression analysis revealed that the odds of anxiety were significantly lower in participants with family-dependent, locally integrated, locally self-contained and wider community-focused networks than in those with a private restricted social network. Further, anxiety was likely to be lower for educated male respondents who were in the higher income quartiles.

## 4. Discussion

This study examined social network typologies and their association with depression and anxiety using a cross-sectional household survey of adults aged ≥30 years in Kerala, India. The study informed that an educated male belonging to higher income quartiles, without any disability, has lower odds of depression and anxiety. Private restricted (28%), locally integrated (26.8%), and wider community-focused (26%) network typologies were the most prevalent in the studied population. Most importantly, the individuals belonging to locally integrated (10.6%) and family-dependent (9.2%), network types had lower levels of depression compared with those belonging to the private restricted (23%) network type. This finding agrees with a population-based cohort study including older people aged ≥65 years in India [29]. Our study also found that among people within a locally integrated network type, only 9% were aged <70, while 20% of people within a private restricted network type were aged above 70. For people aged above 70 years, their higher functional disability acted as a barrier to accessing health-promoting networks and limiting them to private restricted networks. 

A cross-sectional study conducted in eight low-income and developing countries (including India) reported similar findings. People belonging to the private restricted network type had a higher risk of depression than those belonging to integrated social network (family dependent and locally integrated) types [14]. However, observations made in a Western context found out that while people in locally integrated networks have the lowest risk of mental health issues such as depression and loneliness, people within a family-dependent network type have higher odds of depression [30,31]. However, in the current study, the lowest risk of depression was seen in the family-dependent network type. This discrepancy could be explained in terms of the limited number of family connections in the Western context compared to the Eastern. For instance, a person in Kerala looking for familial interactions is more likely to have access to them by virtue of higher average household size (average size of 4.8), compared to small average household sizes in European countries (average size range is from 1.9–2.4) [32]. Further, better financial back-ups and more robust social security systems in the Western context aid in accessing paid online supports, while weaker social security systems and lower wages in India force people to depend mainly on family in case of need [33]. However, further studies in countries with medium household sizes would be required to affirm this association. 

Second to family dependent networks, belonging to a locally integrated network, characterised by social cohesion has the potential to protect people from anxiety or depression. The components of social cohesion: trust, safety and participation [34] can modulate anxiety and depression in the people who share the collectivist value system [35]. Kerala State’s social welfare system has heavily invested in locally integrated networks, introducing a participatory bottom-up people’s planning approach [36]. Most economical and health-related initiatives, such as livelihood promotion and poverty eradication, self-help and neighbourhood groups-based economic engagements, and community health initiatives are navigated through these networks. People belonging to private restricted networks are excluded from many of these social welfare schemes, as most implementations are linked to their participation in these bodies. A cross-sectional study conducted among older people aged ≥60 years in rural Uttar Pradesh, India, demonstrated similar findings in that better social ties with friends and neighbours were protective against depression [37].

Our study also showed a strong association between disability, depression and anxiety. Persons with disabilities had nine times higher depression scores and five times higher anxiety scores than persons with no disabilities. Inadequate access of people with a disability could embed them in restrictive social networks, and lack of integration to a health-promoting network is a detrimental factor to an individual’s mental health. 

A social network can act as a buffer to protect people from depression and anxiety. However, a few other domains modulate adverse outcomes, which require considerations in future studies. For example, the study found that depression was four times more likely among participants with three or more chronic conditions, even if they belonged to family-dependent networks or locally integrated networks. 

Some limitations of this study should be considered while interpreting its findings. Because this is a cross-sectional study, we could not derive any causal inferences from the results. However, this study provides insights into the use of PANT in adults aged above 30 years. Further, this study provides details of the quantitative elements of the networks, such as availability and accessibility to various networks. However, it could not look into the qualitative contribution of these networks in providing help and companionship when people are in need. Further studies are required to establish the qualitative aspects of mental health within these networks in providing help and companionship for the people in need.

## 5. Conclusions

Supportive social networks, especially the family and locally integrated networks, act as buffers to protect people with depressogenic cognitive and social structures from adverse mental health outcomes. Thus, planning and developing comprehensive interventions for people in different social network typologies based on their needs will yield positive health outcomes.

Practitioners can use such network typologies to classify patients into different social network types, which will further assist them in planning and implementing specific social interventions crucial for enhancing patients’ lives. Tailor-made social and psychological interventions can be planned for those in the restricted support networks to effectively manage risk factors. Future research can focus on testing the effectiveness of such interventions as well. Lastly, policymakers can consider the different social network typologies and corresponding co-occurring needs of the patients for their benefit while planning and implementing policies for them. The pivotal role of the family-dependent network in modulating depression and anxiety provides practitioners with the crucial insight that even individual level interventions should have a household focus. Considering family as the most significant source of social support and investing in strengthening the entire household will produce sustainable positive outcomes in the population. 

## Figures and Tables

**Table 1 ijerph-18-06120-t001:** Sociodemographic characteristics by social network typologies.

Variables	Total *N* = 997	Social Network Type					*p* Value *
		Locally Integrated *N* = 267	Local Self-Contained *N* = 97	Wider Community Focused *N* = 256	Family-Dependent *N* = 98	Private Restricted *N* = 279	
Age	53.9 (14.2)	52.7 (12.7)	54.4 (14.2)	52.4 (12.9)	54.1 (15.8)	56.2 (15.9)	F = 3.09, *p* = 0.015
Gender							Χ_2_ = 20.7, *p* < 0.000
Male	365 (36.6%)	125 (46.8%)	28 (28.9%)	94 (36.7%)	36 (36.7%)	82 (29.4%)	
Female	632 (63.4%)	142 (53.2%)	69 (71.1%)	162 (63.3%)	62 (63.3%)	197 (70.6%)	
Income							Χ_2_ = 15.008, *p* = 0.241
1 quartile	417 (41.8%)	109 (40.8%)	40 (41.2%)	98 (38.3%)	49 (50%)	121(43.4%)	
2 quartile	108 (10.8%)	38 (14.2%)	11 (11.3%)	30 (11.7%)	5 (5.1%)	24 (8.6%)	
3 quartile	245 (24.6%)	58 (21.7%)	26 (26.8%)	67 (26.2%)	18 (18.4%)	76 (27.2%)	
4 quartile	227 (22.8%)	62 (23.2%)	20 (20.6%)	61 (23.8%)	26 (26.5%)	58 (20.8%)	
Education							Χ_2_ = 36.9, *p* = 0.002
None	41 (4.1%)	7 (2.6%)	2 (2.1%)	9 (3.5%)	2 (2%)	21 (7.5%)	
Did not complete primary	227 (22.8%)	56 (21%)	26 (26.8%)	40 (15.6%)	22 (22.5%)	83 (29.8%)	
Completed primary	311 (31.2%)	92 (34.5%)	28 (28.9%)	83 (32.4%)	28 (28.6%)	80 (28.7%)	
Completed secondary	216 (21.7%)	62 (23.2%)	17 (17.5%)	61 (23.8%)	23 (23.5%)	53 (19%)	
Completed tertiary	202 (20.3%)	50 (18.7%)	24 (24.7%)	63 (24.6%)	23 (23.5%)	42 (15.1%)	
Religion							Χ_2_ = 17.3, *p* = 0.026
Christian	248 (24.9%)	83 (31.1%)	20 (20.6%)	67 (26.2%)	21 (21.4%)	57 (20.4%)	
Hindu	322 (32.3%)	71 (26.6%)	33 (34%)	89 (34.8%)	41 (41.8%)	88 (31.5%)	
Muslim	427 (42.8%)	113 (42.3%)	44 (45.4%)	100 (39.1%)	36 (36.7%)	134 (48%)	
Disability							Χ_2_ = 79.2, *p* < 0.000
1 st quartile	319 (32.0%)	25 (25.5%)	115 (43.1%)	16 (16.5%)	96 (37.5%)	67 (24.1%)	
2 nd quartile	187 (18.8%)	24 (24.5%)	48 (18%)	23 (23.7%)	56 (21.9%)	36 (12.9%)	
3 rd quartile	257 (25.8%)	28 (28.6%)	60 (22.5%)	30 (30.9%)	68 (26.6%)	71 (25.5%)	
Disabled	234 (23.5%)	21 (21.4%)	44 (16.5%)	28 (28.9%)	36 (14.1%)	105 (37.6%)	
Depression	7.4 (7.3)	6.8 (7.0)	8 (6.9)	6 (5.5)	6.6 (5.9)	9.4 (9.0)	F = 8.92, *p* < 0.000
Anxiety	5.7 (6.7)	4.7 (6.2)	6.0 (6.1)	5.0 (5.9)	5.1 (5.1)	7.6 (8.1)	F = 8.0, *p* < 0.000
Multimorbidity							Χ_2_ = 17.8, *p* = 0.121
No	629(63.1%)	185(69.3%)	63(64.9%)	163(63.7%)	61(62.2%)	157(56.3%)	
One chronic condition	265(26.6%)	61(23%)	26(26.8%)	69(27%)	29(29.6%)	80(28.7%)	
Two chronic condition	85 (8.5%)	17(6.4%)	8(8.3%)	20(7.8%)	5(5.1%)	35(12.5%)	
Two or more than three chronic conditions	18 (1.8%)	4(1.5%)	0	4(1.6%)	3(3.1%)	7(2.5%)	

* One way ANOVA and chi-square tests were used to compare the demographic variables with the different social support networks. Values are numbers (percentage of the number of participants on the variable in question) or means (standard deviation).

**Table 2 ijerph-18-06120-t002:** Association between social network types and depression.

Explanatory Variables	Crude Odds Ratio (95% CI), *p* Value
Social networks	
Private Restricted	1 ref.
Family-Dependent	0.33 (0.16–0.70), *p* = 0.004
Locally integrated	0.56 (0.36–0.87), *p* = 0.010
Local self-contained	0.65 (0.36–1.19), *p* = 0.163
Wider community focused	0.39 (0.24–0.63), *p* = 0.000
Age	1.02 (1.01–1.03), *p* < 0.000
Sex	
Female	1 ref.
Male	0.38 (0.25–0.57), *p* < 0.000
Income	
1st quartile	1 ref.
2nd quartile	0.24 (0.11–0.54), *p* = 0.001
3rd quartile	0.61 (0.39–0.92), *p* = 0.021
4th quartile	0.36(0.22–0.59), *p* < 0.000
Education	
Uneducated	1 ref.
Did not complete primary	0.95 (0.45–1.97), *p* = 0.888
Completed primary	0.38 (0.18–0.80), *p* = 0.011
Completed secondary	0.32 (0.14–0.70), *p* = 0.004
Completed tertiary	0.16 (0.06–0.40), *p* < 0.000
Disability	
Not disabled	1 ref.
Disabled	9.1 (6.2–13.2), *p* < 0.000
Multimorbidity	
No chronic condition	1 ref.
One chronic condition	2.18 (1.47–3.23), *p* < 0.000
Two chronic condition	4.65 (2.78–7.76), *p* < 0.000
Three or more than three chronic conditions	4.27 (1.54–11.74), *p* = 0.005

**Table 3 ijerph-18-06120-t003:** Association between social network types and anxiety.

Explanatory Variables	Crude Odds Ratio (95% CI), *p* Value
Social networks	
Private Restricted	1 ref
Family-Dependent	0.40 (0.22–0.73), *p* = 0.003
Locally integrated	0.40 (0.26 -0.60), *p* = 0.000
Local self-contained	0.50 (0.29–0.88), *p* = 0.016
Wider community focused	0.44 (0.29–0.66), *p* = 0.000
Age	1.01 (1.00–1.02), *p* = 0.05
Sex	
Female	1 ref
Male	0.37 (0.26–0.53), *p* < 0.000
Income	
1st quartile	1 ref
2nd quartile	0.68 (0.40–1.15), *p* = 0.153
3rd quartile	0.65 (0.44–0.96), *p* = 0.031
4th quartile	0.57(0.38–0.86), *p* = 0.007
Education	
Uneducated	1 ref
Did not complete primary	0.64 (0.32–1.28), *p* = 0.207
Completed primary	0.58 (0.29–1.15), *p* = 0.120
Completed secondary	0.36 (0.17–0.74), *p* = 0.006
Completed tertiary	0.22 (0.10–0.48), *p* < 0.000
Disability	
Not disabled	1 ref
Disabled	5.3 (3.8 -7.4), *p* < 0.000
Multi-morbidity	
No chronic condition	1 ref
One chronic condition	2.03 (1.43–2.87), *p* < 0.000
Two chronic condition	4.28 (2.65–6.92), *p* < 0.000
Three or more than three chronic conditions	6.94(2.7–18.03), *p* < 0.000

## Data Availability

The data that support the findings of this study are available from the corresponding author, S.M.D., upon reasonable request.

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
