# Peer review of "Association of Depression and Anxiety with Social Network Types: Results from a Community Cohort Study"

_ijerph, 2021, doi:10.3390/ijerph18116120_

Round 1
Reviewer 1 Report
This is a fine paper that supports the unsurprising conclusion that people with quantitatively better social networks fare better against depression and anxiety than people who have tightly limited social networks. It expands on work done among older adults to include those age 30 and up.
Here are a few things for the authors to consider:
Writing style: please do a very close edit of the writing in the discussion and conclusion. There are numorous small but distracting language mistakes. Running the paper through something like Grammarly should take care of most of them. Please rewrite the sentence on page 7 that begins "Approximately 28%..." so that it is easier for the reader to match the percentage with the network being described. I found myself going back and forth to figure which percentage goes with which situation.
Proximity as a measure of social closeness. I'll take the authors' word on this. It certainly wouldn't work in my context where one way daily commutes of 80km are common. But, it would be nice to have a sentence or two explaining the choice of these particular distances as cut-off points.
I really don't like the reification of Eastern versus Western values and its use as an explanation for the differential importance of family connections. This seems to me to both reinforce old prejudices and efface the vast differences among different communities, nationalities, and ethnicities. I also wonder if there isn't a far simpler explanation: perhaps the main reason why family connections in Europe are less effective than in India is that there are simply fewer of them. A quick check shows the average household size in Kerala as 4.3 but the average household size in Europe as 2.3. So, the person in Kerala who relies on family connections is likely to have access to more of them. European individualism may play some role in this but so might housing markets, average wages, fertility rates, marketing and advertising, and many other factors. It would be interesting to do the same study in China where household size lies between Europe and India.
Reviewer 2 Report
Manuscript deals with the association of several types of social networks and emotional disorders (anxiety and depression). Also, other clinical conditions were assessed (disability level, presence of chronic problems…), in a large community sample. I think this thematic fits part of the scope and aims of IJERPH.
As critical aspects, next I present some comments. I hope they can help to improve this interesting manuscript:
- Authors adopt Wenger’s social network model. I can understand why, but I think a more consistent justification is needed. Social network plays a protective role for mental (and other) disorders, as it plays a social support role. I have not clear Wenger’s model measures satisfaction with existing social network.
- In that sense, anxiety and depression variables are taking as mental health problems. Why only those emotional disorders? One reason is its high prevalence rates. But social support had had been observed as a protective factor at the beginning of the onset of emotional disorders (e.g., doi: 10.3390/ijerph17196984 in this journal), but there are other mental disorders where social network / social support can protect o stress them (e. g., psychotic disorders).
- In method section, participants and ‘procedure’ are not described (because it was done in a previous paper). I think a scientific publication must be understanding by itself. Please, provide at least a general description (including year/s when data were obtained). And, please, explain a little more why 30 years old or olders.
- Can be provided internal consistencies of outcome measures (attained with the study sample)?
- In results section, I primary descriptive data in outcome measures (i.e., means and standard deviations).
- Comparative analysis (for the five social networks) was executed with chi squares test. Two commentaries: (i) chi square is difficult to understand when there are numerous contrasts (e.g., in ‘age’ a unique contrast was done for 25 percentages). (ii) I think an ANOVA (with post hoc contrasts) can be useful with continuous variables (such as depression and anxiety), because this analysis provides precise differences between the five networks (and other independent variables).
- I think data from logistic regressions are insufficient explained. If I am correctly understanding, ODDs ratio of networks (both for anxiety and depression levels) are lower than 0. Does it mean all networks play a ‘protective’ role for anxiety and depression, because there are more ‘no cases’ than ‘cases’? Similar data are collected for income level and education level. Why to be Hindu is ‘protective’ and to be Muslim ‘risk’?
- Minors: the acronym SWADES is used in abstract section, but is defined later (in intro section). Also, please think to move this acronym from the title (acronyms are correctly used when they are widely known). Take care speaking about depression or anxiety participants, because a screening test was used (I think it is better ‘depression scores’ or ‘depression level’). In table 1 are collected (*) undefined symbols. There are changes in letter types, in reference section.
Round 2
Reviewer 2 Report
I think authors have cover the main concerns of my review. Manuscript can be accepted for publication.